# Determinants of Psychosocial Resilience Resources in Obese Pregnant Women with Threatened Preterm Labor—A Cross-Sectional Study

**DOI:** 10.3390/ijerph182010590

**Published:** 2021-10-10

**Authors:** Agnieszka Bień, Ewa Rzońca, Joanna Grzesik-Gąsior, Agnieszka Pieczykolan, Ewa Humeniuk, Małgorzata Michalak, Grażyna Iwanowicz-Palus, Artur Wdowiak

**Affiliations:** 1Chair of Obstetrics Development, Faculty of Health Sciences, Medical University of Lublin, 4-6 Staszica St., 20-081 Lublin, Poland; agnesmbien@gmail.com (A.B.); spupalus@gmail.com (G.I.-P.); 2Department of Obstetrics and Gynecology Didactics, Faculty of Health Sciences, Medical University of Warsaw, 14/16 Litewska St., 00-575 Warsaw, Poland; erzonca@wum.edu.pl; 3Department of Midwifery, Carpathian State College in Krosno, 6 Kazimierza Wielkiego St., 38-400 Krosno, Poland; joanna.grzesik@onet.pl; 4Faculty of Health Sciences, Students’ Scientific Association at the Chair of Obstetrics Development, Medical University of Lublin, 4-6 Staszica St., 20-081 Lublin, Poland; 5Chair and Department of Psychology, Medical University of Lublin, 7 Chodźki St., 20-093 Lublin, Poland; ewa.humeniuk@umlub.pl; 6Department of Gynecology, Independent Public Teaching Hospital No. 4 in Lublin, 8 Jaczewskiego St., 20-954 Lublin, Poland; malgorzata.michalak93@gmail.com; 7Chair of Obstetrics and Gynaecology, Faculty of Health Sciences, Medical University of Lublin, 4-6 Staszica St., 20-081 Lublin, Poland; wdowiakartur@gmail.com

**Keywords:** pregnancy, preterm labor, self-efficacy, life orientation, health locus of control

## Abstract

*Background*: The purpose of the study was to assess the level of such psychosocial resilience resources as self-efficacy, dispositional optimism, and health locus of control in pregnant women with obesity with threatened premature labor. *Methods:* The study was performed in the years 2017–2020 in a group of 328 pregnant women hospitalized due to threatened preterm labor and diagnosed with obesity before the pregnancy. The following instruments were applied: the Life Orientation Test, the Generalized Self-Efficacy Scale, and the Multidimensional Health Locus of Control Scale. *Results:* Obese pregnant women with threatened premature labor have a moderate level of generalized self-efficacy (28.02) and a moderate level of dispositional optimism (16.20). Out of the three health locus of control dimensions, the highest scores were recorded in the “internal control” subscale (26.08). Statistically significant predictors for the self-efficacy variable model included: satisfactory socio-economic standing (ß = 0.156; p = 0.004), being nulliparous (ß = –0.191; p = 0.002), and the absence of comorbidities (ß = –0.145; p = 0.008). Higher levels of dispositional optimism were found in women who were married (ß = 0.381; p = 0.000), reported a satisfactory socio-economic standing (ß = 0.137; p = 0.005), were between 23 and 27 weeks pregnant (ß = –0.231; p = 0.000), and had no comorbidities (ß = –0.129; p = 0.009). *Conclusions:* Generalized self-efficacy in obese women with threatened preterm labor is associated with satisfactory socio-economic standing, being nulliparous, and the absence of chronic disease. Dispositional optimism in obese pregnant women with threatened preterm labor is determined by their marital status, socio-economic standing, gestational age, and the absence of comorbidities.

## 1. Introduction

Despite improvements in healthcare quality, there is an upward trend in the incidence of preterm deliveries (before 37 weeks of pregnancy). The number of children born prematurely each year is estimated at 15 million. The percentage of preterm births ranges between 5% and 18% of all births. Due to its multi-factor etiology, the problem remains a challenge in modern medicine and perinatal care, as it is associated with increased neonatal morbidity and mortality [1,2]. The risk of premature labor is increased e.g., in obese women. The association between obesity and preterm labor results from the increased risk of pregnancy complications in obese women, compared to those with a normal body weight [3]. In recent years, the prevalence of obesity (defined as a BMI > 30 kg/m^2^) in women has increased considerably. Approximately 8% of women are now classified as severely (morbidly) obese, i.e., have a BMI ≥ 40 kg/m^2^. According to estimates, the global prevalence of obesity in women is expected to exceed 21% by 2025 [4,5]. During pregnancy, maternal obesity poses risks both to the mother and the fetus, and significantly increases health care costs due to its association with a variety of obstetric conditions, including preterm labor [3,6].

Threatened preterm labor is a challenge for the woman, who may experience strong emotions, such as anxiety, sadness, or guilt. Stress levels and ways of coping with stress in these circumstances depend on the structure of the woman’s personality, the support she receives, and her psychosocial resilience resources [7]. Psychosocial resilience resources are intrapersonal and social characteristics of an individual that favor constructive responses to life’s requirements and minimize the impact of any stressors experienced. These resources include self-efficacy, dispositional optimism, and health locus of control [8,9]. Self-efficacy is an individual’s belief in their ability to achieve their goal through their own effort and perseverance. It is a determinant of health behaviors, it assists in coping with stress and pain, it facilitates the understanding of others’ behaviors, and even stimulates immune system function [10]. During pregnancy and after delivery, it plays a major role in a woman’s adaptation to the difficult and challenging role of a mother [11,12]. Dispositional optimism is a tendency to believe one will experience positive events in one’s life. In difficult situations, it favors more active ways of coping [13]. Health locus of control refers to the belief that one has a real impact on one’s health [14]. 

Pregnancy is a period of intense physiological and psychological change, entailing a number of adaptive processes. If the pregnancy is perceived by the woman as a threatening event, her adaptive mechanisms may be disrupted. This is certainly the case when pregnancy complications occur, including threatened preterm labor [15].

### Purpose of the Study

The purpose of the study was to evaluate self-efficacy, life orientation, and health locus of control (psychosocial resilience resources) in pregnant women with obesity and threatened premature labor, and to identify the determinants of these variables.

## 2. Materials and Methods

### 2.1. Study Groups 

The study was performed in the years 2017–2020 in a group of 328 pregnant women, hospitalized due to threatened preterm labor and diagnosed with obesity before the pregnancy in accordance with the World Health Organization’s international classification, i.e., with a BMI equal to or exceeding 30 kg/m^2^ [16]. 

Gestational age was identified based on the patients’ medical records. In the study, the category of preterm labor was broken down into: extremely preterm (before 28 weeks), very preterm (28 to 32 weeks), and moderate to late preterm (32 to 37 weeks) [17]. A survey questionnaire was administered to each respondent on the last day of her hospitalization. Due to the scope of the study, the sample was selected in a targeted, rather than probabilistic manner. Inclusion criteria for the study group of pregnant patients were as follows: consent to participate in the study, age above 18 years (the legal age of majority in Poland), hospitalization at a high-risk pregnancy ward, gestational age between 22 and 37 weeks, obesity, Caucasian race, speaking Polish as one’s native language, singleton pregnancy, and receiving proper prenatal care since the beginning of the pregnancy. The exclusion criteria were: unconfirmed gestational age, multiple pregnancy, or diagnosis of a lethal fetal anomaly (Figure 1).

Table 1 shows the participants’ characteristics. In the study group, most respondents were aged 26–35 years (57.9%), urban residents (54.6%), married/in a stable relationship (72.3%), had not completed higher education (51.2%), and had a satisfactory socio-economic standing (51.8%); typically, they were pregnant for the second time (44.5%), nulliparous (76.5%), between weeks 32 and 37 (36.3%), and had chronic diseases (54.6%). 

The study was approved by the Lublin Medical University Bioethics Committee (approval no. KE-0254/284/2017). Each participant was informed about the purpose of the study and provided with questionnaire completion instructions. Respondents were informed that participation was voluntary, and that the study results were anonymous and to be used exclusively for research purposes. All respondents provided their informed consent in writing. Out of the 360 survey questionnaires distributed to respondents, 328 correctly completed questionnaires were analyzed, and the data effectiveness rate was 91.11%.

### 2.2. Assessments

The study used a diagnostic survey with questionnaires. The following instruments were applied: The Life Orientation Test (LOT-R), the Generalized Self-Efficacy Scale (GSES), the Multidimensional Health Locus of Control Scale (MHLC), and a standardized interview questionnaire with items concerning the participants’ characteristics.

The Generalized Self-Efficacy Scale (GSES) evaluates an individual’s value system manifested in their ability to cope with difficult situations. The scale comprises 10 statements rated on a scale of 1 to 4 (1—disagree, 2—somewhat disagree, 3—somewhat agree, 4—agree). The total score reflects the overall level of self-efficacy, with higher scores indicating more self-efficacy. Scores between 10 and 24 points are interpreted as a low level of self-efficacy, 25–29 points—moderate, and 30–40 points—a high level of self-efficacy. Cronbach’s alpha for scale reliability is 0.85, and the internal consistency of the GSES ranges between 0.76 and 0.91 [18,19]. 

The Revised Life Orientation Test (LOT-R) evaluates the respondent’s dispositional optimism based on 10 statements, six of which are diagnostic, while four are filler items. Each is rated on a five-item scale, from 0—strongly disagree, to 4—strongly agree. Total scores range between 0 and 24 points. Scores of 17–24 points indicate a high level of dispositional optimism, 13–16—moderate optimism, and 0–12—a pessimistic disposition. Cronbach’s alpha for the scale’s internal consistency is 0.76 [20].

The Multidimensional Health Locus of Control Scale (MHLC) comprises 18 statements rated on a six-item scale, which represent convictions referring to generalized expectations in three health locus of control dimensions: internal factors (I am in control of my health), external factors/impact of others (my health results from the actions of others, including medical personnel), and the belief that one’s health results from random events. The total score for each subscale is between 6 and 36 points. Higher scores indicate a stronger belief that the factor of interest affects one’s health. Scores are interpreted based on the median value: those above the median are considered high, and those below the median are considered low. Scale reliability is 0.64 for internal control, 0.59 for impact of others, and 0.63 for random events [21].

### 2.3. Statistical Analysis

Statistical analysis of the material collected was performed using Statistica software, version 12.5. In the descriptive analysis, means and SD values, frequencies, and percentages were calculated. Stepwise regression was used to identify predictors of LOT-R, GSES, and MHLC scores. Stepwise regression is a method of regression model fitting in which the choice of predictive variables is performed using an automatic procedure. For the purpose of this analysis, dummy coding was used for variables such as residence, education, or support from loved ones. Correlations between quantitative variables were calculated. Linearity assumptions and variance homogeneity were tested with scatter plots, and there was no heteroscedasticity. Multicollinearity was measured by the variance inflation factor (VIF). For the goodness of model fit, adjusted R-squared and overall F-test were considered. P-values below 0.05 were used to identify independently associated factors in multivariate linear regressions.

## 3. Results

Table 2 reports the mean scores among the women with threatened preterm labor for generalized self-efficacy (28.02 ± 3.67), dispositional optimism (16.20 ± 3.95), and health locus of control (MHLC) broken down into internal factors (26.08 ± 3.68), impact of others (21.52 ± 4.06), and belief that one’s health is determined by random events (19.08 ± 5.36).

Our statistical analysis demonstrated significant positive correlations between the respondents’ sense of generalized self-efficacy on the one hand and their dispositional optimism and internal health locus of control on the other, and between dispositional optimism and internal health locus of control. The correlations were rated at between 0.129 and 0.479. There were also negative correlations between the women’s generalized self-efficacy and their health locus of control in the external factors and random events dimensions, as well as between attribution of health locus of control to external factors and to random events. The strength of correlations was between −0.434 and −0.120 (Table 3).

Table 4 reports regression analysis results for generalized self-efficacy (GSES), and dispositional optimism (LOT-R) scores in the women studied. Statistically significant predictors for the self-efficacy variable model included: satisfactory socio-economic standing (*ß* = 0.156; *p* = 0.004), being nulliparous (*ß* = –0.191; *p* = 0.002), and the absence of comorbidities (*ß* = –0.145; *p* = 0.008). Multilevel variable scanning showed higher levels of dispositional optimism in women who were married (*ß* = 0.381; *p* = 0.000), reported a satisfactory socio-economic standing (*ß* = 0.137; *p* = 0.005), were between 23 and 27 weeks pregnant (*ß* = –0.231; *p* = 0.000), and had no chronic comorbidities (*ß* = –0.129; *p* = 0.009).

The regression model for the health locus of control (MHLC) variable is shown in Table 5. External locus of control was positively associated with being married (*ß* = 0.115; *p* = 0.040), having a satisfactory socio-economic standing (*ß* = 0.121; *p* = 0.030), and having given birth at least once before (*ß* = 0.124; *p* = 0.044). Higher scores for the “random events” locus of control variable were recorded for women who were single (ß = –0.281; *p* = 0.0001), had an unsatisfactory socio-economic standing (*ß* = –0159; *p* = 0.002), were 32 weeks pregnant (*ß* = 0.227; *p* = 0.000), and lived in rural areas (*ß* = 0.115; *p* = 0.027). In the case of the internal health locus of control variable, the proposed regression model had a poor fit to the data (F = 1.692; *p* = 0.090).

## 4. Discussion

Pregnant women with obesity have a higher incidence of obstetric complications such as miscarriage, pregnancy-induced hypertension, preeclampsia, and eclampsia [22,23,24,25]. In addition, maternal obesity is associated with a higher risk of preterm labor, which, according to Slack et al. [26], increases along with the degree of obesity. Threatened preterm labor is a major challenge for a woman, which causes chronic stress and fear for oneself and one’s baby, and requires hospitalization. An individual’s way of coping with difficulties and their perception of stress are influenced, among other factors, by their psychosocial resilience resources [7].

The purpose of the present study was to assess the level of such psychosocial resilience resources as self-efficacy, dispositional optimism, and health locus of control in pregnant women with obesity and threatened premature labor, as well as the determinants of these resources.

Self-efficacy relies on a cognitive process whereby an individual evaluates their ability to handle a variety of situations. Research to date indicates that it is an important predictor of the attitudes, emotions, and behaviors of pregnant women [27,28]. However, the literature on the subject is limited. The present study broadens the understanding of the topic and is among the first ever to analyze generalized sense of self-efficacy (GSES scores) among obese pregnant women with threatened premature labor. The mean GSES score was 28.02, within the upper limits of the mean reference value range. Similar findings were reported in studies on primigravid women in the third trimester of pregnancy (28.29) [29], in pregnant women with hyperglycemia (31.58) [30], and in women who had miscarried (30.29) [31].

Significant GSES predictors in the present study included: a satisfactory socio-economic standing, being nulliparous, and the absence of chronic diseases. In the study by Brunton et al. [32], self-efficacy among mothers was correlated with acceptance of pregnancy, but, as in the present study, was uncorrelated with age. Soh et al. [33] found higher self-efficacy in multiparous women reporting better psychological wellbeing, and lower in those who had delivered by cesarean section and had more labor-related anxiety. In others studies, self-efficacy was also reported as a predictor of such health-related behaviors in pregnant women as avoidance of second-hand smoke [34] or good oral hygiene [35], but also of concern for the child and attitudes towards medical personnel and towards labor [32]. Women who have delivered healthy children at term and have a high level of self-efficacy tend to view their parental competence more favorably and be more satisfied with perinatal care [11,12].

A moderate level of self-efficacy may enable women experiencing a threat to pregnancy to assess their situation accurately and seek effective ways of dealing with the difficulties and obstacles they encounter—in this case, the risk of a preterm delivery and its potential consequences. They may also be expected to be more involved in the treatment process, showing better compliance and adherence, e.g., by resting more, avoiding exertion and stress, and abstaining from substance use.

Dispositional optimism is a tendency to believe one will experience positive events in one’s life. Optimistically disposed individuals demonstrate more active coping strategies, lower levels of psychological stress, positive health-related behaviors, and better physical functioning, among other characteristics [36]. In our study, the pregnant women with threatened premature labor had a moderate level of dispositional optimism. Moyer et al. [37] studied dispositional optimism and health-related quality of life in pregnant women and found higher levels of optimism in respondents who were better educated, professionally active, and without pregnancy complications. Loh et al. [38] found that higher levels of optimism were correlated with positive health-related behaviors, lower levels of parental stress, and better quality of life in the mothers studied.

In our study, being married was significantly associated with dispositional optimism. Patients who are supported by their partners tend to perceive their own future in a more positive manner, even in a difficult situation, such as that of hospitalization. In this context, the study by Giangiordano et al. [13] is also of interest, as the authors found higher levels of dispositional optimism in pregnant women to be associated with such variables as age (30 or above), being in a relationship (married or in a stable informal relationship), and education (high school or higher).

We found an association between dispositional optimism in pregnant women with threatened premature labor and earlier gestational week. Notably, the need to be hospitalized during pregnancy represents a major challenge for a woman. During hospitalization, pregnant patients undergo frequent examinations, have to remain immobilized during infusions or cardiotocography, and may be placed on bed rest. In addition, their wellbeing may be impaired by medication, e.g., tocolytics. These patients live in constant fear and uncertainty as to whether all these treatments and sacrifices will allow them to achieve their goal, i.e., give birth to a healthy baby. Dispositional optimism moderates a factual assessment of the situation, and increases the woman’s motivation, perseverance, and determination.

Another psychosocial resilience resource analyzed in the present study was health locus of control. Individuals with an internal locus of control have a sense of control over their own health and are thus likely to take specific measures to improve or maintain it. In turn, those with an external locus of control believe that, regardless of their own actions, things are decided by external factors. Therefore, they are convinced that health-promoting behaviors are ineffectual, and thus tend not to engage in actions to maintain or improve their health. Health locus of control may also be attributed to random events. In this case, one has no sense of control over their health, which interferes with health-promoting behaviors [19,39]. Out of all health locus of control dimensions, the highest scores were obtained on the internal control subscale. The tendency to place the health locus of control internally is desirable in pregnant women, as a conviction about being in control of one’s health promotes positive health-related behaviors (such as proper nutrition, avoidance of substance use, reduction/avoidance of stress, hygienic lifestyle, regular medical check-ups), thus limiting the risk factors for various pregnancy complications [39,40]. Kordi et al. [41] demonstrated a positive correlation between taking control of one’s own health and self-care activities. 

Threatened preterm labor is a stressful situation which entails a number of challenges for a woman, and therefore entrusting control of one’s health to others, including medical personnel, may support treatment aimed at prolonging the pregnancy and limiting the consequences of preterm birth. Our findings indicate that the perception of one’s health as dependent on external factors is more common among women who are married, satisfied with their living and financial situation, and nulliparous. Women who attribute their health to the impact of others are more likely to rely on the opinion of specialists and seek sources of support to help them cope with the situation. This is corroborated by Kordi et al., [41] who demonstrated that pregnant women with gestational diabetes mellitus attributed control over their health mostly to others. This seems positive as all pregnancy complications require the patient to comply with recommendations from the medical staff.

In the group of pregnant women with obesity that was studied, the conviction that one’s health is dependent on random events was more common in those who had an unsatisfactory socio-economic standing and lived in rural areas. Patients with such an outlook tend to be more passive in their health-related behaviors, and are not particularly consistent in adhering to medical recommendations or undergoing regular diagnostics. This may be assumed to result from a lack of consistency in adhering to recommendations and undergoing regular diagnostics in patients who are not convinced of their ability to actively modify their own health [39,40,41]. As these women’s motivation is low, they are not proactive in their health-related behaviors and are unlikely to follow professional advice on health; therefore, these patients will require longer and more intensive health education interventions.

Care for women with obesity and threatened preterm labor should include interventions to reinforce their self-efficacy, which fosters adaptation to one’s current health situation. The appropriate management of pregnant patients should not only focus on health or economic benefits—it should also include an evaluation of the patients’ psychosocial resilience resources that favor constructive coping with one’s life situation and its requirements, and minimize the impact of stressors the patients encounter. Our study is the first that we know of to analyze self-efficacy, life orientation, and health locus of control in pregnant women with obesity diagnosed with threatened premature labor. Notably, we used standardized instruments, and so other researchers interested in issues related to premature labor will be able to compare results, continue in-depth research, and draw conclusions. The results obtained here may help clarify the importance of care going beyond professional medical interventions and including strategies of holistic care for pregnant women with threatened premature labor. The appropriate behaviors of medical personnel, education, and support may all contribute to the optimization of obstetric care and positively affect the psycho-physical condition of women with threatened premature labor.

## 5. Conclusions

Obese women with threatened premature labor have a moderate level of generalized self-efficacy, a moderate level of dispositional optimism, and an internal health locus of control. 

Generalized self-efficacy in obese women with threatened preterm labor is associated with a satisfactory socio-economic standing, being nulliparous, and the absence of chronic disease.

Dispositional optimism in obese pregnant women with threatened preterm labor is determined by their marital status, socio-economic standing, gestational age, and the absence of comorbidities.

Being married, having a satisfactory socio-economic standing, and having given birth at least once before are factors positively associated with an external health locus of control, while a locus of control attributed to random events is determined by the woman’s marital status, socio-economic standing, residence, and gestational age.

## Figures and Tables

**Figure 1 ijerph-18-10590-f001:**
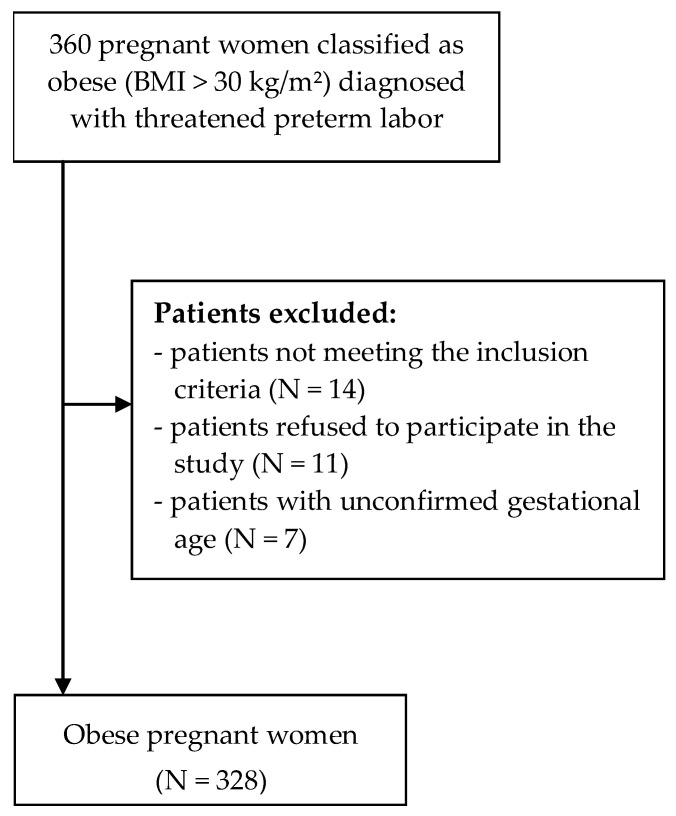
Flowchart of the recruitment process of the patients.

**Table 1 ijerph-18-10590-t001:** Characteristics of women in the study.

Characteristics of the Group		N	%
Age	18–25 y/o	95	29.0
	26–35 y/o	190	57.9
	More than 35 y/o	43	13.1
Residence	Urban	179	54.6
	Rural	149	45.4
Relationship status	Married	237	72.3
	Single	91	27.7
Education	Other than higher	168	51.2
	Higher	160	48.8
Socio-economic standing	Satisfying	170	51.8
	Not satisfying	158	48.2
Number of pregnancies	First pregnancy	132	40.2
	Second pregnancy	146	44.5
	Third or subsequent pregnancy	50	15.2
Number of previous deliveries	None	251	76.5
	One	72	22.0
	Two or more	5	1.5
Week of pregnancy	23-27 Hbd	96	29.3
	28-32 Hbd	113	34.5
	32-37 Hbd	119	36.3
Concurrent chronic disease: hypertension, diabetes, thyroid and heart diseases	No	149	45.4
	Yes	179	54.6

**Table 2 ijerph-18-10590-t002:** Generalized self-efficacy, dispositional optimism, and health locus of control scores in obese pregnant women with threatened preterm labor.

Resilience Resources		M	Me	SD	Min	Max
GSES		28.02	28.00	3.67	15.00	38.00
LOT R		16.20	17.00	3.95	4.00	24.00
MHLC	Internal	26.08	27.00	3.68	15.00	36.00
	Impact of others	21.52	22.00	4.06	10.00	31.00
	Random events	19.08	19.00	5.36	6.00	32.00

GSES—Generalized Self-Efficacy Scale; LOT-R—Life Orientation Test–Revised; MHLC—Multidimensional Health Locus of Control Scale.

**Table 3 ijerph-18-10590-t003:** Correlations between GSES, LOT-R, and MHLC scores in pregnant women with obesity and threatened preterm labor.

		GSES	LOT-R	MHLC		
				Internal	Impact of Others	Random Events
GSES		-				
LOT-R		0.479 **	-			
MHLC	Internal	0.365 **	0.129 *	-		
	Impact of others	−0.149 **	0.062	0.099	-	
	Random events	−0.120 *	−0.434 **	−0.032	−0.125 *	-

GSES—Generalized Self-Efficacy Scale; LOT-R—Life Orientation Test–Revised; MHLC—Multidimensional Health Locus of Control Scale. * *p* < 0.05; ** *p* < 0.01.

**Table 4 ijerph-18-10590-t004:** Regression analysis results for GSES and LOT-R scores in obese pregnant women with threatened preterm labor.

Predictors	GSES F = 3.888; *p* < 0.001; R^2^ = 0.074					LOT-R F = 12.890; *p* < 0.001; R^2^ = 0.247				
	*B*	*SE*	*β*	*t*	*p*	*B*	*SE*	*β*	*t*	*p*
Age	−0.033	0.314	−0.006	−0.106	0.916	0.433	0.304	0.069	1.422	0.156
Residence ^A^	0.660	0.403	0.090	1.639	0.102	−0.023	0.390	−0.003	−0.059	0.953
Relationship status ^B^	0.734	0.445	0.090	1.648	0.100	3.349	0.431	0.381	7.763	0.000
Socio-economic standing ^C^	1.146	0.399	0.156	2.876	0.004	1.084	0.386	0.137	2.808	0.005
Education ^D^	0.014	0.402	0.002	0.036	0.971	0.130	0.389	0.016	0.333	0.739
Number of pregnancies ^E^	0.720	0.454	0.096	1.586	0.114	−0.514	0.440	−0.064	−1.169	0.243
Number of previous deliveries ^F^	−1.650	0.520	−0.191	−3.174	0.002	−0.267	0.503	−0.029	−0.530	0.597
Week of pregnancy	−0.364	0.250	−0.080	−1.460	0.145	−1.129	0.242	−0.231	−4.669	0.000
Occurrence of chronic diseases:^G^	−1.068	0.402	−0.145	−2.658	0.008	−1.020	0.389	−0.129	−2.621	0.009

GSES—Generalized Self-Efficacy Scale; LOT-R—Life Orientation Test–Revised; *β*—standardized coefficients. *SE*—bootstrapped standard errors. Reference categories: ^A^ residence—rural; ^B^ married; ^C^ satisfactory socio-economic standing; ^D^ higher education; ^E^ second or subsequent pregnancy; ^F^ at least one previous delivery; ^G^ chronic disease.

**Table 5 ijerph-18-10590-t005:** Regression analysis results for MHLC scores in pregnant women with threatened preterm labor.

Predictors	MHLC—Impact of Others F = 2.258; *p* = 0.018; R^2^ = 0.033					MHLC—Random Events F = 7.986; *p* < 0.001; R^2^ = 0.161				
	*B*	*SE*	*β*	*t*	*p*	*B*	*SE*	*β*	*t*	*p*
Age	−0.028	0.355	−0.004	−0.079	0.937	−0.433	0.436	−0.051	−0.992	0.322
Residence ^A^	−0.256	0.455	−0.031	−0.562	0.575	1.240	0.559	0.115	2.216	0.027
Relationship status ^B^	1.038	0.503	0.115	2.064	0.040	−3.359	0.619	−0.281	−5.431	0.000
Socio-economic standing ^C^	0.984	0.450	0.121	2.186	0.030	−1.708	0.554	−0.159	−3.085	0.002
Education ^D^	0.270	0.454	0.033	0.594	0.553	0.002	0.558	0.000	0.004	0.997
Number of pregnancies ^E^	0.230	0.513	0.028	0.448	0.654	−0.204	0.630	−0.019	−0.323	0.747
Number of previous deliveries ^F^	1.189	0.587	0.124	2.024	0.044	−0.995	0.722	−0.079	−1.378	0.169
Week of pregnancy	0.074	0.282	0.015	0.263	0.793	1.506	0.347	0.227	4.344	0.000
Occurrence of chronic diseases: ^G^	0.572	0.454	0.070	1.259	0.209	0.710	0.558	0.066	1.271	0.205

MHLC—Multidimensional Health Locus of Control Scale; *β*—standardized coefficients. *SE*—bootstrapped standard errors. Reference categories: ^A^ residence—rural; ^B^ married; ^C^ satisfactory socio-economic standing; ^D^ higher education; ^E^ second or subsequent pregnancy; ^F^ at least one previous delivery; ^G^ chronic disease.

## Data Availability

The datasets generated during and/or analysed during the current study are available from the corresponding author on reasonable request.

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
