# Peer review of "Determinants of Psychosocial Resilience Resources in Obese Pregnant Women with Threatened Preterm Labor—A Cross-Sectional Study"

_ijerph, 2021, doi:10.3390/ijerph182010590_

Round 1

Reviewer 1 Report

Page 3: Fig.1: The last part of the figure should be unified:

                 Obese pregnant women...         Normal weight pregnant women...

Page 4-5, r. 163-175 (incl. Table 1) should be placed in the chapter "material and methods" just under the Fig. 1. From the Table 1 it is evident, that the gropus are not matched in socio-economic standing and comorbidities what may influence the results. This should be corrested. the groups are large enough to do it.

Page 5 Table 2: what is the "Cohena"?

Page 6 Table 3: It should be mentioned if the table refers to both the groups.

                         Should be probably - ** p < 0,01

Page 6-7: Table 4 refers to the obese patients. It should be useful to present analogical table for control group. What is "liczba porodów"?

Page 7: Table 5 refers probably to both the groups - obese and control but it should be mentioned. It should be useful to create two tables to compare obese and control patients. What is "liczba porodów"?

Page 7- 13: The discussion is generally too long.

Page 13: Conclusions: As there are two groups of pregnant women (obese and control) it should be clearly visible in conclusions - e.g. obese women compared to non-obese have higher/lower level of...

Author Response

Dear Reviewer,

The Authors would like to thank the Reviewer for many valid remarks, comments and suggestions. We have revised our paper accordingly and as requested We have provided below a detailed response to reviewer’ comments.

Page 3: Fig.1: The last part of the figure should be unified:

Obese pregnant women...         Normal weight pregnant women...

We have changed figures descriptions.

Page 4-5, r. 163-175 (incl. Table 1) should be placed in the chapter "material and methods" just under the Fig. 1. From the Table 1 it is evident, that the gropus are not matched in socio-economic standing and comorbidities what may influence the results. This should be corrested. the groups are large enough to do it.

We have changed table descriptions to make them more readable. Based on the reviews, we came to the conclusion that considering the study concept, design, and purpose — to evaluate self-efficacy, life orientation, and health locus of control in obese pregnant women with threatened preterm labor — it is best to only leave in the study group and remove the controls, as all analyses performed only concerned the obese women with threatened preterm labor.

Page 5 Table 2: what is the "Cohena"?

It was our mistake – we have changed table descriptions.

Page 6 Table 3: It should be mentioned if the table refers to both the groups.

We have changed table descriptions from which it follows, that results in table 3 refers only to study group – obese pregnant women with threatened preterm labor.

Should be probably - ** p < 0,01

Good point, we have changed descriptions.

Page 6-7: Table 4 refers to the obese patients. It should be useful to present analogical table for control group. What is "liczba porodów"?

We have changed those records.

Page 7: Table 5 refers probably to both the groups - obese and control but it should be mentioned. It should be useful to create two tables to compare obese and control patients.

The purpose of the study was to evaluate self-efficacy, life orientation and health locus of control (psychosocial resilience resources) in pregnant women with obesity and threatened premature labor, and to identify the determinants of these variables, that why regression analysis in table 4 and 5 refers only to study group – obese pregnant women with threatened preterm labor.

What is "liczba porodów"?

We have changed these records.

Page 7- 13: The discussion is generally too long.

According to above suggestion the discussion has been shortened.

Page 13: Conclusions: As there are two groups of pregnant women (obese and control) it should be clearly visible in conclusions - e.g. obese women compared to non-obese have higher/lower level of...

We have modified the conclusions to match the current results, with the controls removed. The Reviewer's suggestions have been implemented and highlighted in blue.

We are grateful for the feedback and the opportunity to respond to the Reviews’ comments and suggestions. We hope the changes we have made improve the overall quality of the paper in line with your expectations.

Reviewer 2 Report

The paper is timely and helps to put in a different perspective an important and often misconstrued health-related problem. Despite this, some issues undermine my interest. I suggest to the authors to carefully check the entire paper searching for typos and to emend the text from the reading issue, in particular adopting for all the work the same terminology, often not-so-consistent through the entire manuscript. In particular, the introduction had to be improved to strengthen the link between actual research and the rationale, in some parts the introduction and rationale sections seem to be disconnected from the following parts of the present work.  

Author Response

Dear Reviewer,

The Authors would like to thank the Reviewer for many valid remarks, comments and suggestions. We have revised our paper accordingly and as requested We have provided below a detailed response to reviewer’ comments.

The paper is timely and helps to put in a different perspective an important and often misconstrued health-related problem. Despite this, some issues undermine my interest. I suggest to the authors to carefully check the entire paper searching for typos and to emend the text from the reading issue, in particular adopting for all the work the same terminology, often not-so-consistent through the entire manuscript. In particular, the introduction had to be improved to strengthen the link between actual research and the rationale, in some parts the introduction and rationale sections seem to be disconnected from the following parts of the present work.  

As suggested, we have harmonized the terminology used, improved the Introduction section, and revised the manuscript in terms of language.

The Reviewer's suggestions have been implemented and highlighted in blue.

We are grateful for the feedback and the opportunity to respond to the Reviews’ comments and suggestions. We hope the changes we have made improve the overall quality of the paper in line with your expectations.

Round 2

Reviewer 2 Report

The paper seems improved for real, I suggest only to check carefully for minor language improvements